# The Effects of Cryogenic Storage on Human Dental Pulp Stem Cells

**DOI:** 10.3390/ijms22094432

**Published:** 2021-04-23

**Authors:** Nela Pilbauerova, Jan Schmidt, Tomas Soukup, Romana Koberova Ivancakova, Jakub Suchanek

**Affiliations:** 1Department of Dentistry, Charles University, Faculty of Medicine in Hradec Kralove and University Hospital Hradec Kralove, 500 05 Hradec Kralove, Czech Republic; nela.pilbauerova@lfhk.cuni.cz (N.P.); jan.schmidt@lfhk.cuni.cz (J.S.); suchanekj@lfhk.cuni.cz (J.S.); 2Department of Histology and Embryology, Faculty of Medicine in Hradec Kralove, Charles University, 500 03 Hradec Kralove, Czech Republic; soukupto@lfhk.cuni.cz

**Keywords:** dental stem cells, cryopreservation, uncontrolled-rate freezing, stem cell storage, regenerative medicine

## Abstract

Dental pulp stem cells (DPSCs) are a type of easily accessible adult mesenchymal stem cell. Due to their ease of access, DPSCs show great promise in regenerative medicine. However, the tooth extractions from which DPSCs can be obtained are usually performed at a period of life when donors would have no therapeutic need of them. For this reason, it is imperative that successful stem cell storage techniques are employed so that these cells remain viable for future use. Any such techniques must result in high post-thaw stem cell recovery without compromising stemness, proliferation, or multipotency. Uncontrolled-rate freezing is not a technically or financially demanding technique compared to expensive and laborious controlled-rate freezing techniques. This study was aimed at observing the effect of uncontrolled-rate freezing on DPSCs stored for 6 and 12 months. Dimethyl sulfoxide at a concentration of 10% was used as a cryoprotective agent. Various features such as shape, proliferation capacity, phenotype, and multipotency were studied after DPSC thawing. The DPSCs did not compromise their stemness, viability, proliferation, or differentiating capabilities, even after one year of cryopreservation at −80 °C. After thawing, they retained their stemness markers and low-level expression of hematopoietic markers. We observed a size reduction in recovery DPSCs after one year of storage. This observation indicates that DPSCs can be successfully used in potential clinical applications, even after a year of uncontrolled cryopreservation.

## 1. Introduction

Dental pulp stem cells (DPSCs) have a crucial advantage over other adult stem cells. They are easily accessible from pulp tissues. As extracted teeth are typically liquidated as biological waste, it is very convenient to preserve these teeth as a source of DPSC isolation. Dental pulp tissues from impacted or semi-impacted wisdom teeth are most frequently used as a source of DPSCs [1]. Surgical removal of impacted mandibular third molars should be carried out well before the age of 24 years. Older patients are at higher risk of postoperative complications [2]. The second most frequent source of DPSCs involves premolars [1]. They are mostly extracted during orthodontic treatment of severe tooth crowding, class II malocclusion [3]. The patient’s age is usually around 10 to 16 years. Supernumerary teeth, especially in the midline (mesiodens), are the third most frequent DPSC isolation source [1]. Removal of the mesiodens before the age of five is associated with fewer complications and a reduced need for potential orthodontic treatment in the future [4]. As mentioned earlier, extractions are usually performed at a period of life when donors/patients would not need their isolated dental pulp stem cells. Therefore, the question is how to store the isolated DPSCs with no effect on their stemness features until such a time when donors/patients might need them in regenerative or reparative therapies.

Cryopreservation is a process of sustaining cell and tissue viability by freezing and storing at sub-zero temperatures, during which biochemical reactions do not occur [5]. However, stored cells are exposed to stressful conditions during cryopreservation, which might lead to irreversible damage or cell death (so-called cryoinjury). Cryobiology is a multidisciplinary science that studies the physical and biological behavior of cells at sub-zero temperatures (lower than the freezing point of water) [6]. It is the process involved in the freezing of water that can be particularly problematic in cryopreservation. Freezing injuries are caused either by the formation of ice crystals, by cell destruction via direct mechanical action, or through secondary effects brought about by changes in the solute concentration or osmotic homeostasis changes extra- or intracellularly [7]. Depending on whether the freezing rate is slow or fast, intracellular water either flows across the cell membrane and joins the extracellular ice phase or freezes intracellularly and forms ice crystals inside the cells [8]. Cryoprotectants (CPs) are incorporated in a cryopreservation medium in order to obtain higher cryosurvival of stored cells. There are two categories of CPs, distinguished according to their molecular weight. Low-molecular-weight CPs (LMW-CPs) include dimethyl sulfoxide (DMSO), glycerol, and ethylene (propylene) glycol [7]. These can penetrate a cell cytoplasmic membrane, preventing ice crystal nuclei formation and slowing down the rate of ice crystal growth inside cells [9]. The second group, high-molecular-weight CPs (HMW-CPs), includes dextran, hydroxyethyl starch, polyvinyl-pyrrolidone, and polyvinyl alcohol. These remain in the extracellular space and participate in cell dehydration, thus minimizing intracellular ice crystal formation but helping in membrane stabilization [10]. However, HMW-CPs are not enough on their own and are often used in combination with LMW-CPs in order to reduce the LMW-CPA concentration. The most versatile CP is the dimethyl sulfoxide (DMSO), commonly at a concentration of 10%. The adverse effects of DMSO have been known since the first clinical trials in the 1960s. A DMSO concentration of 10% or below is generally accepted as non-toxic but concerns still remain. Therefore, it would be beneficial to find an alternative CP with the same or similar protective effects in cryopreservation but without concerns over toxicity.

Cryopreservation protocols might also vary in the freezing technique used. There are several cryopreservation protocols: controlled-rate freezing, uncontrolled-rate freezing, rapid freezing (vitrification), and magnetic freezing technique [11]. The vitrification process is applied to a high CP concentration, and rapid freezing causes the glass-like solidification of cells and tissues [12]. This method is mainly used for the preservation of sperms and oocytes [13]. A controlled rate can minimize the cell damage caused by rapid freezing and ice crystal formation. The manual or programmable controlled-rate freezers precisely control the freezing rate of 1–2 °C per minute or 0.3–0.6 °C per minute [14]. During magnetic freezing, the freezers also apply a weak magnetic field to block water molecules from clustering, thus preventing the formation of ice crystals [15]. The uncontrolled-rate freezing is not a technically or financially demanding technique compared to the previously mentioned methods. Uncontrolled-rate freezing has been most frequently applied for the preservation of peripheral blood stem [16,17,18,19]. Only a few published studies have described the uncontrolled-rate freezing during subsequent storage of stem cells at −80 ° C [20,21]. In all available studies so far, the effect of uncontrolled-rate cryopreservation on DPSCs has been observed using the DMSO. Woods et al. reported that the optimal number of cells for cryopreservation is 1.0–1.5 × 10^6^ cells, using 10% DMSO as the CP [22].

The purpose of this study was to cryopreserve DPSCs for 6 and 12 months using the uncontrolled-rate freezing, and observe the effects on cell size, viability, proliferation activity, phenotype, and differentiation potential of stored stem cells. We used 10% DMSO as a cryoprotective agent.

## 2. Results

Cryopreservation Effect on DPSC Characteristics

The study was aimed at observing the effect of uncontrolled-rate freezing on DPSC features such as cell morphology, viability, proliferation rate, phenotype, and differentiation potential. To carry out such observation, we successfully isolated ten lineages of DPSCs and characterized them during their cultivation (up to the 8th passage). These data were used as a control group (DPSC-control). The 1.5 × 10^6^ DPSCs from each lineage were harvested in the first passage and stored using uncontrolled-rate freezing for 6- and 12-month periods. For cell retrieval after cryostorage, we used a 37 °C thermal bath. After thawing, we followed the same cultivation protocol we set for the negative control.

The dominant shape of the non-cryopreserved DPSCs were fibroblast-like with long cytoplasmic fibers. We did not observe any cell morphology changes after a 6-month or 12-month cryopreservation (Figure 1).

The average size of non-cryopreserved cells (DPSC-control) was 14.2 ± 0.5 µm. The average size of cells cryopreserved for 6 months (DPSC-6M) was 13.9 ± 0.6 µm, and the size of cells cryopreserved for 12 months (DPSC-12M) was 13.7 ± 0.8 µm (Figure 2). The average size of cryopreserved cells was smaller compared to control samples. The difference was statistically significant in the case of DPSC-12M (* *p* > 0.05).

We also evaluated the effect of uncontrolled-rate freezing on the viability of DPSCs. We tested the viability based on the trypan blue extrusion method. The average percentage of viable DPSC-6M was 88.6 ± 4.3% in the second passage and 93.3 ± 3.2% in the eighth passage. The DPSC-12M figures were 91.3 ± 4.3% in the second passage and 88.8 ± 6.2% in the eighth passage. In comparison with the negative control (92.2 ± 1.8% in the second passage and 92.7 ± 2% in the eighth passage), the data significantly differed only in the viability of DPSC-6M measured in the second passage (*p* < 0.01) (Figure 3.)

One of the remarkable features of stem cells, including DPSCs, is their proliferation capacity. DPSCS, like other adult stem cells, have the ability to self-renew. This characteristic plays a crucial role in the potential usage of cells in regenerative or reparative medicine. All groups of cells in our study remained proliferatively active until the eighth passage (Figure 4). The DPSC-control reached 47.2 ± 3.2 PDs from the primary passage (passage immediately after stem cell isolation) to the eighth passage. DPSC-6M reached almost the same number of cumulative PD (47.7 ± 2.7), while the PD number of DPSC-12M was lower than the one for fresh samples (44.6 ± 4.1), a figure that differed significantly (*p* < 0.01). We also observed a statistically significant difference in the fifth and seventh passages (*p* < 0.05). The DPSC-6M figures were not statistically different in any of the passages compared to the control group.

We observed an increasing trend in PDT with the increasing number of passages. Cryopreservation did not change this trend. The average PDT for DPSC-control, DPSC-6M, and DPSC-12M was 55.1 ± 24.1 h, 61.6 ± 21.1 h, and 58.8 ± 30.6 h, respectively (Figure 5). The PDT of cells stored for 12 months in the sixth and eighth passages varied with statistical significance from those in the control group (*p* < 0.01).

Phenotype profiles of DPSCs were analyzed using flow cytometry in the third passage (3p) and seventh passage (7p). The DPSC-control showed high positivity (>71%, [23]) for CD13, CD29, CD44, CD73, CD90, and for CD166, which are the surface markers for mesenchymal stem cells or stromal associated surface markers. The cells showed low (<41%, [23]) or no expression (<11%, [23]) of CD10, CD18, CD31 (the endothelial marker), CD34, and CD45 (hematopoietic markers), CD271, STRO-1 and MHC class II. The DPSC-control showed moderate expression of CD105 and MHC class I (<70%, [23]). Cryopreserved cells (DPSC-6M and DPSC-12M) showed significantly lower or higher expression of CD13, CD29, and CD 90 in selected passages compared to the control group. However, expression of these markers remained over 71%. We observed a significantly higher expression of endothelial marker CD31 after cryopreservation than in the control group (<11%). We calculated no significant changes in the negative expression of the hematopoietic marker CD34. We did not observe any significant changes in expression of CD10, CD63, CD105, MHC class II. The expression of CD18 was significantly higher in DPSC-12M in the third passage, but median remained below 10%. Conversely, the expression of MHC class I was significantly lower in DPSC-6M in the third passage and STRO-1 in DPSC-12M in the seventh passage compared to DPSC-control. However, medians of both markers remained in the same range as the figures for DPSC-control. Expressions of all evaluated surface markers in the control group, DPSC-6M and DPSC-12M, are shown in the layouts of graphs (Figure 6). We also observed a significantly lower expression of CD117 and higher expression of CD106 and CD146 in cryopreserved cells than in fresh cells.

We also studied the effect of the uncontrolled-rate freezing on DPSC multipotency at six- and twelve-month storage at −80 °C. Multipotency is one of the main features of stem cells, primarily in their potential usage in regenerative or reparative therapies. It was observed that cryopreserved cells, even after 12-months storage, can differentiate into mature cell populations. We were able to trigger osteogenesis and chondrogenesis. We confirmed our observation using histological staining and immunocytochemistry (Figure 7, Figure 8, Figure 9 and Figure 10).

Despite the standard adipogenic differentiation medium, fresh samples differentiated in adipocytes unwillingly. The accumulated adipose vacuoles or droplets were scarcely apparent in the DPSC-control group (Figure 11). Due to this observation in fresh samples, we did not study the cryopreserved cells’ ability to differentiate in the adipose cell line.

## 3. Discussion

After first being identified by Gronthos and colleagues in 2000 [24], DPSCs have become the subject of much scientific research due to their similarities with mesenchymal stem cells. This similarity manifests not only in their shared fibroblast-like morphology (with selective adherence to plastic surfaces) but also in their good proliferative potential and ability to differentiate into multiple cell lineages in vitro. DPSCs have been shown to differentiate into other cell types (including all three germ layers): odontogenic, osteogenic, chondrogenic, neurogenic, myogenic, endothelial cell lines, and cells producing insulin [25,26]. The broad spectrum of differentiation potential and relative ease of harvest favor DPSCs for use in research related to cell therapy and regenerative medicine. The initial results of in vitro research indicate the potential future use in the treatment of endocrine disorders (diabetes mellitus), neurodegenerative diseases (Parkinson’s disease, Alzheimer’s disease, stroke), spinal injury, peripheral nerve injury, and in the repair and regeneration of bones and cartilage (osteoporosis, osteoarthritis) [27].

Cryopreservation offers several benefits. It eliminates the need to keep cells in long-term culture and the consequent problems (such as risk of contamination, genetic drifts, or epigenetic changes). Cryopreservation also allows the maintenance of a particular cell phenotype of stored cells in cell banks, and enables cells to be stored for future research, as well as for clinical use and testing; it also simplifies cell transport between individual facilities. However, it is necessary to understand the cryopreservation process to apply it fully and optimize the individual steps, thus minimizing the adverse freezing effects on stored cells. In cryopreservation, the main goal is to preserve the structural and functional integrity of cells after thawing. To achieve this goal, it is necessary to introduce a cryoprotectant into the cryopreservation medium. The most widely used and versatile CP is DMSO, especially at a concentration of 10%. The relatively high dose (880 g applied on skin or 320 g injected intravenously would result in 50% mortality in 80 kg humans [28]) brought about a change in how DMSO toxicity is viewed, with the U.S. Food and Drug Administration classifying DMSO in the same class as ethanol, namely a class 3 solvent [29]—the safest category with low toxic potential at levels commonly accepted in pharmaceuticals. Nowadays, DMSO is used as a solvent in toxicology and pharmacology for the cryopreservation of a broad spectrum of cells, and as a penetration enhancer during topological treatments.

Standardized freezing protocols and procedures must adhere to good manufacturing practice (cGMP) to ensure that cryopreserved cells are potentially suitable for clinical application. Due to stem cell diversity, it is unlikely that a universal cryopreservation protocol can be achieved. The optimum cryopreservation protocol for DPSC banking should be straightforward and effective. Uncontrolled-rate freezing is not a technically or financially demanding method. This method has been most frequently applied to preserve peripheral blood stem cells with consistent results [16,17,18,19]. Kumar et al. studied various uncontrolled-rate freezing protocols and noted that samples directly stored at −80 °C in 10% DMSO in uncontrolled temperature conditions for a month showed the highest proliferation rate [20].

DPSCs can survive for extended periods and can be passaged several times. Our research team’s previous study showed that DPSCs achieved 60 population doublings in culture medium, with no spontaneous differentiation [30]. Laino and co-workers accomplished 80 passages by maintaining the DPSCs substrate interaction and cell-cell communication in the secreted extracellular matrix central region [31]. However, extensive in vitro proliferation of human DPSCs is associated with telomere attrition, with a significant correlation to prolonging DPSC population doubling time. The telomere length is determined as a biomarker which indicates the replicative age of stem cells [32,33]. Cryopreservation eliminates the need to keep cells in long-term culture.

The goal of our study was to evaluate the effect of the uncontrolled-rate and 10% DMSO as CP on DPSCs stored for 6 and 12 months. We evaluated the effect on DPSC morphology, viability, phenotype, proliferation activity, and differentiation potential in subsequent passages after thawing. We successfully isolated ten lineages of DPSCs from permanent teeth to achieve this goal. The spectrum of the source teeth generally corresponds with the trend described in other studies [1]. Third molars were harvested the most, followed by premolars, with mesiodens being the third most frequent.

After thawing the DPSCs in a 37 °C thermal bath, we did not observe changes in cell morphology. The fibroblast-like shape with long cytoplasmic fibers remained the dominant shape even after the 12 months of cell storage. The average size of cryopreserved cells was smaller compared to control samples. The DPSC-12M size reduction was statistically significant. This can be explained when we consider that stress during cryopreservation and thawing conditions result in selecting and eliminating various subpopulations of DPSCs. Generally speaking, cells with a higher volume and surface area are more affected by the osmotic stress resulting from ice crystal formation, and more easily lyse during freezing or thawing conditions [6].

Another problem associated with cryopreservation is cell death after thawing. In our study, we evaluated the viability in subsequent passages after thawing in order to determine the recovery of DPSC after thawing. After one year of storage, the viability of DPSCs was over 88% at the end of the cultivation, which is still higher than has been described in other studies [20,21]. We observed a viability reduction in DPSC-6M in the second passage. This drop was statistically significant compared to control samples. However, the viability of DPSC-6M was over 93% in the eighth passage. As we did not observe this phenomenon in the DPSC-12M, we hypothesize that technical difficulties might cause this viability reduction during CP addition or removal. DMSO toxicity increases with higher temperature, concentration, and exposure time [34]. Therefore, we assume that DPSCs-6M were exposed to DMSO at a higher temperature for a prolonged period of time.

Non-cryopreserved cells experienced a prolongation of population doubling time with increased passaging. Cryopreserved cells copied this trend. Although DPSC-12M reached a significantly lower PD number for higher PDT in the eighth passage, both groups of cryopreserved cells remained proliferative active. DPSC proliferation potential is one of the main features of stem cells, especially in their potential usage in cell-based therapies. In general, cryopreserved cells needed more time for post-thaw recovery to regain their “functional” health.

It has been demonstrated that DPSCs are heterogeneous cells with several populations significantly varying in many biological features. The minimal criteria set by The International Society for Cellular Therapy assures MSC identity by using CD70, CD90, and CD105 as positive markers, and CD34 as a negative marker. However, the three positive markers are co-expressed in a wide variety of cells, and, therefore, even when used in combination, they are certainly incapable of identifying MSCs in vivo [35]. We analyzed a broad spectrum of CD markers in fresh and cryopreserved samples. Even though we observed some significant changes, cryopreserved cells kept highly expressing (<71%) CD markers for mesenchymal stem cells: CD29, CD44, CD90, and stromal associated markers CD13, CD73, and CD166. We observed a statistically significant decline in the expression of CD90 and CD13 in cryopreserved cells and an increase in the expression of CD29 in DPSC-12M. The hematopoietic stem cell marker (CD34) remained in low-level expression, even after 12-months storage. The low positivity of CD45 can be explained by the cultivation of DPSCs in a medium enriched with ITS, which keeps cells at a less differentiated level [36]. We also observed a significantly higher expression of surface marker CD31 (PECAM-1, platelet endothelial cell adhesion molecule-1) in the DPSC-12M. This increase can be explained by the action of free oxygen radicals [37] or in immature dental pulp stem cells [38]. It has been demonstrated that free oxygen radicals are also formed during the cryopreservation process [39]. Another surface molecule typical for endothelial cells, CD106 (VCAM-1, vascular cell adhesion protein 1), was also expressed significantly higher in DPSC-6M in the third passage. This surface molecule can also be activated due to free oxygen radicals, mainly during inflammatory response [40].

The expression of CD146 (MCAM, melanoma cell adhesion molecule) was upregulated after 6-months and 12-months storage. CD146 has been seen as a marker for mesenchymal stem cells isolated from multiple adult and fetal organs, and its expression may be linked to multipotency [41,42]. A high variability of CD146 has also been observed and published in many studies [43,44]. We do not suggest that cryopreservation would change the ability of DPSC to differentiate, but it might cause the selection of DPSC subpopulations with upregulated surface marker CD146. Controversially, the surface marker CD117 (tyrosine-protein kinase KIT) was downregulated in cryopreserved samples. CD117 plays a role in cell survival, proliferation, and differentiation [45,46]. However, we studied multipotency in cryopreserved DPSCs and, due to the results, DPSCs were able to differentiate into chondroblast and osteoblast cell lines. We did not trigger adipogenesis in cryopreserved cells because, when we tried to induce it in non-cryopreserved samples, DPSCs differentiated in adipocytes unwillingly. This result is not surprising, since adipocytes are not a physiological component of dental pulp tissues. These observations correspond with the Gronthos study [24].

## 4. Materials and Methods

### 4.1. DPSC Lineages

We successfully isolated 10 dental pulp stem cell lineages from donors aged 13 to 18 (Table 1). All donors or their legal representatives were informed about the ongoing study before they signed the informed consent. University Hospital Hradec Kralove’s ethical committee approved the study guidelines and the informed consent content (ref. no. 201812 SO7P). DPSCs cryopreserved for 6 months were indicated by the letter A, and cells cryopreserved for 12 months were indicated by the letter B.

Teeth were extracted at the Dental Clinic at University Hospital Hradec Kralove. We have already described the DPSCs isolation technique in our previous study [11]. Briefly, the DPSC isolation was performed on the same day as the tooth extraction. We isolated DPSCs using an enzymatic isolation technique using a 0.5% trypsin/EDTA solution (Gibco, UK) for 10 minutes. Before the enzymatic digestion, the minced dental pulp tissues were ground using a homogenization method to obtain fine and homogenous pieces of pulp tissues, thus facilitating the enzyme effect and shortening the digestion period. After one week of cultivation, we observed small cell colonies adhering to a cultivation dish. We then removed the cultivation medium to wash out the remaining non-adherent cells, parts of extracellular mass, and vessels. The DPSCs were cultivated in a modified cultivation media Minimum Essential Medium Eagle—alpha modification; (Alpha-MEM, Gibco, Gaithersburg, MD, USA) for mesenchymal adult progenitor cells containing 2% fetal bovine serum (FBS, PAA Laboratories, Toronto, ON, Canada), and supplemented with 10 ng/mL epidermal growth factor (PeproTech, London, UK), 10 ng/mL platelet-derived growth factor (PeproTech), 50 mM dexamethasone (Sigma-Aldrich, St. Louis, MO, USA), 0.2 mM L-ascorbic acid (Bieffe Medital) for protection against oxygen radicals, essential amino acid glutamine (Invitrogen, Carlsbad, CA, USA) at a final concentration of 2%, and antibiotics—100 U/mL penicillin, 100 µg/mL streptomycin (Invitrogen), 20 μg/mL gentamicin (Invitrogen), and 0.4 µl/mL amphotericin (Sigma-Aldrich, St. Louis, MO, USA). The medium was also enriched with 10 μL/mL Insulin-Transferrin-Selenium-Sodium supplement (ITS, Bieffe Medital) to increase the nutrient utilization. We kept the cultivation dishes at a temperature of 37 °C and at 5% CO_2_. The cultivation medium was changed every three days and, after the cells reached a 70% confluence, we passaged them at a final concentration of 5000 cells/cm^2^. We ended the cultivation when cells reached the 8th passage.

### 4.2. The Uncontrolled-Rate Freezing

The cells were stored using the uncontrolled-rate freezing for 6 and 12 months. We cryopreserved 2 cryovials containing 0.5 milliliters of medium with 1.5 × 10^6^ cells isolated from the 1st passage. The cryopreservation medium composed of FBS and DMSO (Sigma-Aldrich–Merck KGaA, Darmstadt, Germany) as the CP was cooled down to a temperature of 4 °C and then mixed with cell pellet. The final DMSO concentration was 10% after mixing, with the cells immersed in the cultivation medium. The cryovials were placed at a temperature of –20 °C and kept for 1–1.5 h. Then they were placed directly in the freezer and stored at –80 °C for 6 and 12 months.

### 4.3. Cell Thawing

After the 6-months storage, one cryovial from each sample (lineage) was thawed using a 37 °C warm thermal bath. The cells were mixed in a tube with an inactivation medium (α-MEM and FBS at a concentration of 20%) to inactivate the cryopreservation medium and the cryoprotectant. The tube was then centrifuged at 2000 rpm for 5 min (600g). First, the obtained pellet was resuspended and then seeded in the new cultivation dish. Subsequently, we continued in the same cultivation protocol as with the negative control. We proceeded the same way with the second cryovial from each sample (lineage) after the 12-months storage.

### 4.4. Cryopreservation Effect on DPSC Characteristics

In order to reach the primary goal of this study, we observed the effect of uncontrolled-rate freezing on the underlying biological features of DPSCs. The lineages of non-cryopreserved cells were taken as the control. The cell count and cell diameter were measured using Z2-Counter (Beckman Coulter, Miami, FL, USA). The proliferation activity measurement is described in a previous study [33]. Briefly, the proliferation activity was determined as cumulative population doublings (PDs) and population doubling time (PDT). We used the formula PD = log_2_ (N_x_/N_1_) to calculate the population doublings reached in each passage. N_x_ is the total passage cell count calculated using the Z2-Counter, and N_1_ is the initial cell count seeded into the culture dish (5000 cells/cm^2^). To calculate the population doubling time, we used the formula PDT = t / n, where t is the number of hours of cultivation per passage and n is the number of PDs in that passage, calculated as described above.

We assessed the cell viability using a trypan dye exclusion method in the 2nd and 8th passage using the Vi-Cell analyzer (Beckman Coulter, USA). The trypan dye did not penetrate viable cells due to their intact membrane. According to the number of cumulative population doublings (PD) and population doubling time (PDT), proliferation capacity was measured. Cryopreserved cells were thawed in the 1st passage. In order to calculate the number of PD and PDT, we took the values reached in the primary and 1st passage from the negative control.

We also wanted to determine the cryopreservation effect on the DPSC phenotype. We repeated 21 cluster of differentiation (CD) marker analyses using a flow cytometer Cell Lab Quanta (Beckman Coulter) in the 3rd and 7th passages. First, we detached adherent stem cells using the 0.05% trypsin-EDTA solution (Gibco, London, UK); we then stained them with primary immunofluorescence antibodies conjugated with phycoerythrin (PE) or fluorescein (FITC) against the following CD markers: CD10 (CB-CALLA, eBioscience, San Diego, CA, USA), CD13 (WM-15, eBioscience, San Diego, CA, USA), CD18 (7E4, Beckman Coulter, Brea, CA, USA), CD29 (TS2/16, BioLegend, San Diego, CA, USA), CD31 (MBC 78.2, Invitrogen, Carlsbad, CA, USA), CD34 (581 (Class 287 III), Invitrogen, Carlsbad, CA, USA), CD44 (MEM 85, Invitrogen, Carlsbad, CA, USA), CD45 (HI30, Invitrogen, Carlsbad, CA, USA), CD49f (GoH3, Invitrogen, Carlsbad, CA, USA), CD63 (CLBGran/12, Beckman Coulter, Brea, CA, USA), CD73 (AD2, BD Biosciences 288 Pharmingen, Erembodegen, Belgium), CD90 (F15-42-1-5, Beckman Coulter, Brea, CA, USA), CD105 (SN6, 289, Invitrogen), CD106 (STA, BioLegend, San Diego, CA, USA), CD117 (2B8, Chemicon, Tokyo, Japan), CD146 (TEA1/34, Beckman Coulter, Brea, CA, USA), CD166 (3A6, Beckman Coulter, Brea, CA, USA), CD271 (ME20.4, BioLegend, San Diego, CA, USA), MHC class I (Tu149, Invitrogen, Carlsbad, CA, USA), MHC class II (Tü36, Invitrogen, Carlsbad, CA, USA), and STRO-1 (STRO-1, BioLegend, San Diego, CA, USA). Positive cells were determined as the percentage with a fluorescence intensity greater than 99.5% of the negative isotype immunoglobulin control. An overview of the analyzed phenotype panel is described in Table 2.

One of the features of DPSCs is their multipotency; therefore, we examined the effect of cryopreservation on differential potential as well. We triggered osteogenesis, chondrogenesis, and adipogenesis in cells harvested from the fourth passage. We followed the same differentiation protocols with all groups.

For osteogenic differentiation, the cells were exposed to the Differentiation of Basal Medium-Osteogenic (Lonza, Basel, Switzerland) for three weeks. At the end of differentiation, we examined the origin of the produced extracellular matrix using immunohistochemistry and histological staining. Osteogenic mass with stem cells harvested after three weeks was fixed using 10% formalin, embedded in paraffin, and cut at 7 µm slices. After their deparaffinization, the samples were stained using von Kossa histological staining in order to reveal calcium phosphate deposits as black spots. Immunocytochemistry was used to visualize the main bone protein osteocalcin as rusty colored areas. Samples were exposed to the primary antibody, a primary mouse IgG antibody (1:50, Millipore, Burlington, MA, USA), and a donkey anti-mouse secondary IgG antibody (1:250, Jackson ImmunoResearch Labs).

Chondrogenesis was initiated using the Differentiation Basal Medium-Chondrogenic (Lonza), supplemented with 50 ng/mL TGF-β1 (R&D Systems, Minneapolis, MN, USA). We exchanged the medium every three days for three weeks. Before histological staining, the samples were fixed, paraffined, and stained using blue Masson trichrome to reveal procollagen and collagen in the produced extracellular mass. The collagen and procollagen appeared blue. We also confirmed the presence of collagen type II using a primary mouse IgM antibody (1:500, Sigma-Aldrich, St. Louis, MA, USA) and Cy3TM-conjugated goat anti-mouse secondary IgM antibody. Cell nuclei were counterstained with 4’-6-diamidino-2-phenylindole (DAPI, Sigma-Aldrich, St. Louis, MA, USA).

Differentiation in adipocytes was induced in the monolayer when DPSCs reached 100% confluence with hMSC Adipogenic Induction SingleQuots (Lonza, Basel, Switzerland), and maintained with hMSC Adipogenic Maintenance SingleQuots (Lonza, Basel, Switzerland). Media were used subsequently and switched every three days for three weeks. After that time, cultures were fixed with 10% formalin and stained with oil red (lipid particles vacuoles or droplets stained red).

### 4.5. Statistical Analysis

All statistical analyses were performed using the statistical software GraphPad Prism 6 (San Diego, CA, USA). The data are presented as the mean ± SD or median, with boxes and whiskers representing the interquartile range and 5th–95th percentiles, respectively, according to the data distribution. The statistical significances (* *p* < 0.05, ** *p* < 0.01, *** *p* < 0.001, **** *p* < 0.0001) were calculated using either one-way ANOVA, followed by Dunnett’s multiple comparison test for continuous variables, or Friedman’s test, followed by the Dunn’s multiple comparison test on ranks for nonparametric variables. The Shapiro–Wilk test or Kolmogorov–Smirnov test were used for normal distribution evaluations.

## 5. Conclusions

The study was aimed at evaluating the effect of uncontrolled-rate freezing and 10% DMSO as CP on DPSCs stored for 6 and 12 months. DPSCs were able to withstand stress conditions when they were stored for 12 months, while maintaining the characteristics that typically define mesenchymal stem cells. DPSCs did not compromise their stemness, proliferation, or differentiating capabilities, even after one year of cryopreservation at −80 °C. DPSCs can thus be used as an ideal source for stem cell banking. However, we used DMSO at a concentration of 10% as the cryoprotective agent. Even though 10% concentration is generally accepted for the cryopreservation of stem cells, we would like to lower the DMSO concentration or, ideally, eliminate DMSO from the cryopreservation medium in our future research.

## Figures and Tables

**Figure 1 ijms-22-04432-f001:**
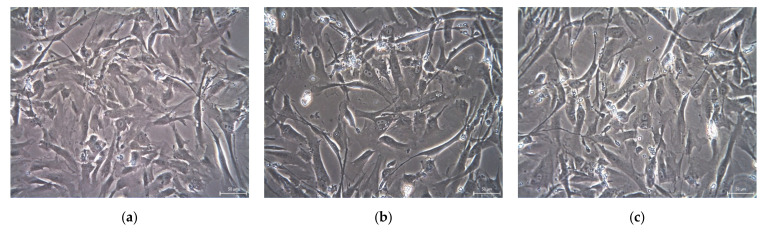
The DPSC morphology (lineage Z02 in the 3rd passage). Scale bar 50 µm. (**a**) DPSC-control; (**b**) DPSC-6M; (**c**) DPSC-12M.

**Figure 2 ijms-22-04432-f002:**
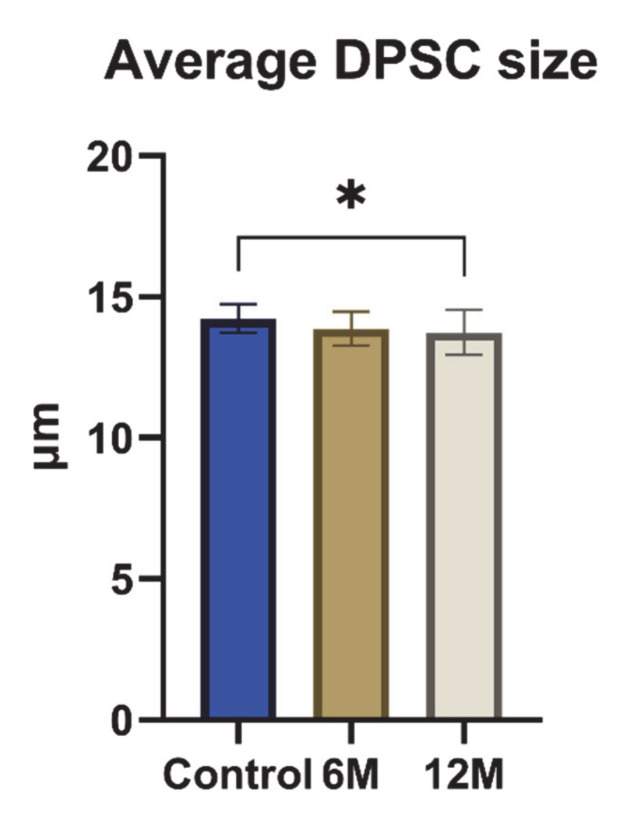
The average cell size of all cell groups in µm during entire cultivation. Data are presented as a mean and SD plotted as error bars. The statistical analysis was performed between fresh samples and cryopreserved samples using one-way ANOVA, followed by Dunnett’s multiple comparison test (* *p* < 0.05).

**Figure 3 ijms-22-04432-f003:**
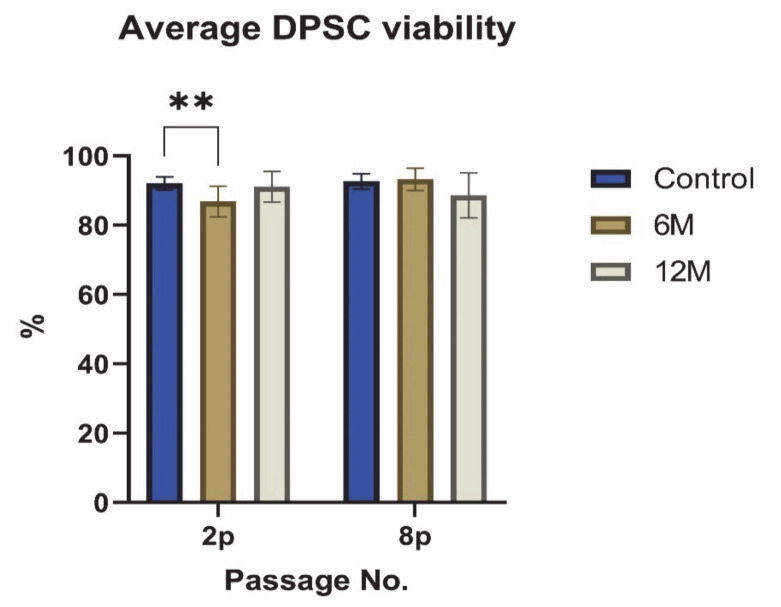
The average DPSC viability of all cell groups evaluated in the 2nd and 8th passages. The graph presents the percentage of viable cells. Data are presented as a mean and SD plotted as error bars. The statistical analysis was performed between fresh samples and cryopreserved samples using one-way ANOVA, followed by Dunnett’s multiple comparison test (** *p* < 0.01).

**Figure 4 ijms-22-04432-f004:**
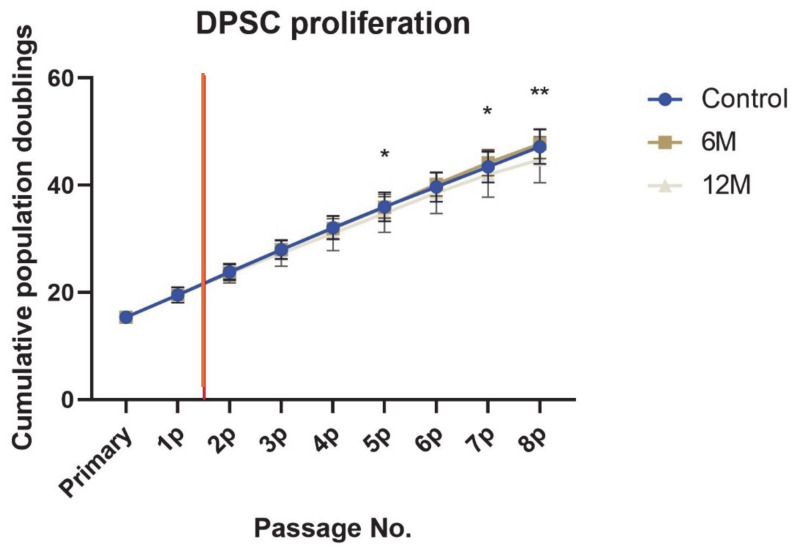
Cumulative population doublings of all cell groups reached from the primary to 8th passage. Data for primary and 1st passage are the same for all cell groups. The red vertical line illustrates the time of cryopreservation and thawing. Data are presented as a mean and SD plotted as error bars. The statistical significance was calculated between fresh samples and cryopreserved samples using either one-way ANOVA, followed by Dunnett’s multiple comparison test for continuous variables, or Friedman’s test, followed by Dunn’s multiple comparison test on ranks for nonparametric variables (* *p* < 0.05), (** *p* < 0.01).

**Figure 5 ijms-22-04432-f005:**
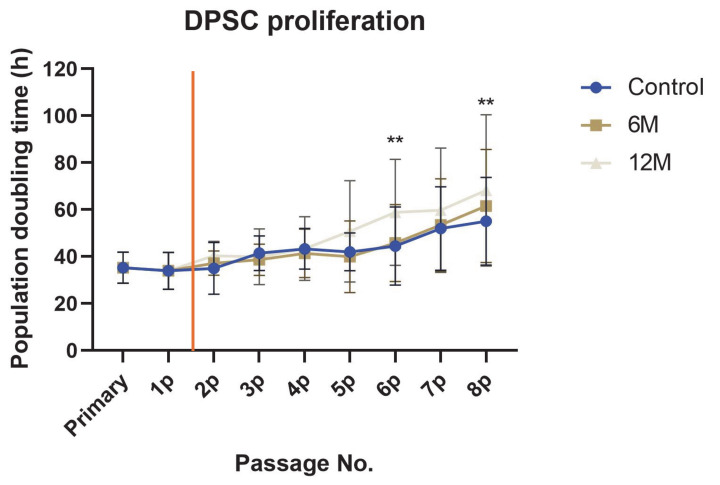
The population doubling time of all cell groups reached from the primary to 8th passage. Data for primary and 1st passage are the same for all cell groups. The red vertical line illustrates the time of cryopreservation and thawing. Data are presented as a mean and SD plotted as error bars. The statistical significances were calculated between fresh samples and cryopreserved samples using either one-way ANOVA, followed by Dunnett’s multiple comparison test for continuous variables, or Friedman’s test, followed by the Dunn’s multiple comparison test on ranks for nonparametric variables (** *p* < 0.01).

**Figure 6 ijms-22-04432-f006:**
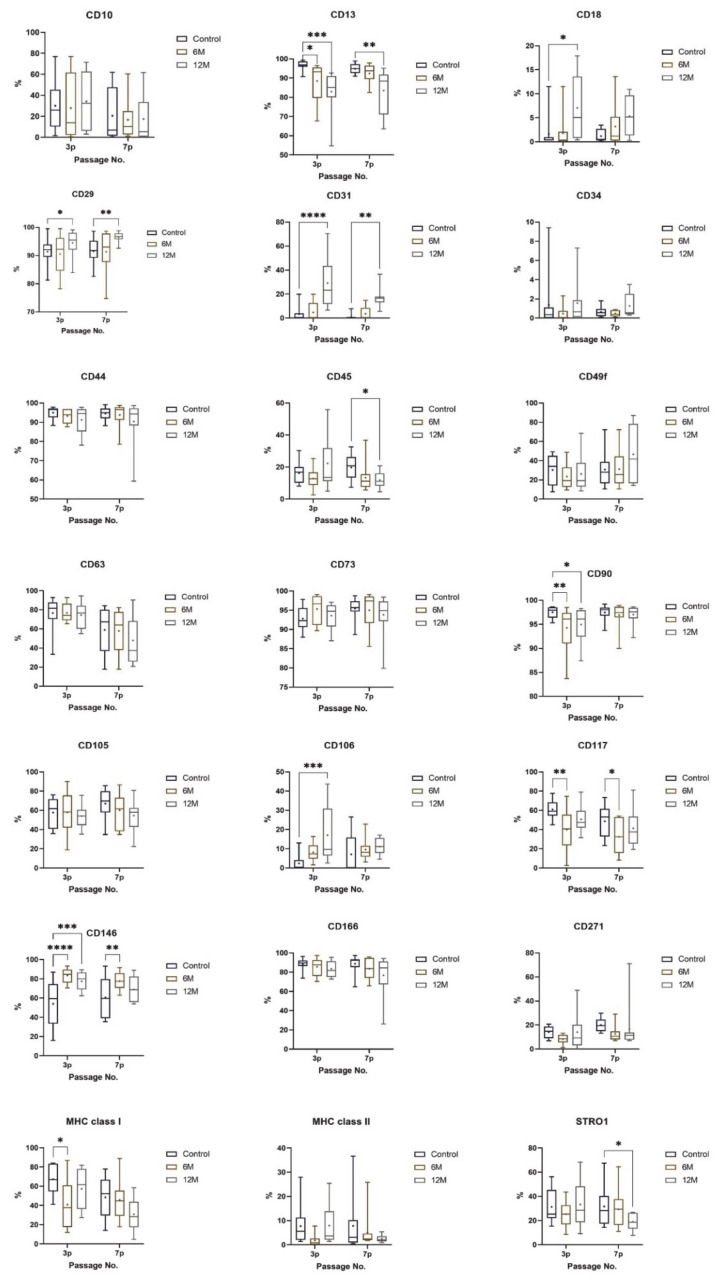
Phenotype profile of all cell group analyses in the 3rd and 7th passages. The layouts of graphs illustrate percentages of positive cells, determined as the percentage with a fluorescence intensity greater than 99.5% of the negative isotype immunoglobulin control. Data are presented as a median, with boxes and whiskers representing the interquartile range and 5th–95th percentiles. The mean is shown as ‘+.’ The statistical comparison was performed between fresh samples and cryopreserved samples using Friedman’s test, followed by Dunn’s multiple comparison test (* *p* < 0.05), (** *p* < 0.01), (*** *p* < 0.001), (**** *p* < 0.0001).

**Figure 7 ijms-22-04432-f007:**
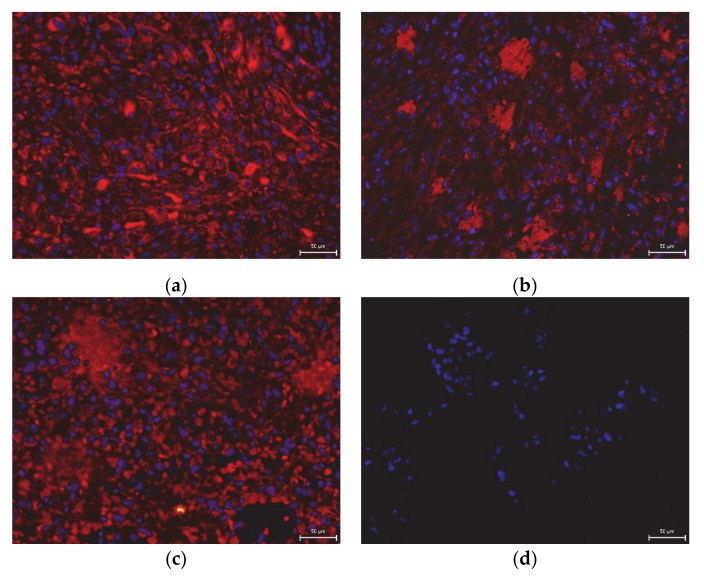
Immunocytochemical detection of collagen type II in the extracellular chondrogenic mass of the lineage Z01 after 3-week cultivation in chondrogenic differentiation medium. The collagen type II fluorescents red, and stem cell nuclei fluorescent blue. Scale bar 50 µm. (**a**) DPSC-control; (**b**) DPSC-6M; (**c**) DPSC-12M; (**d**) non-differentiated cells.

**Figure 8 ijms-22-04432-f008:**
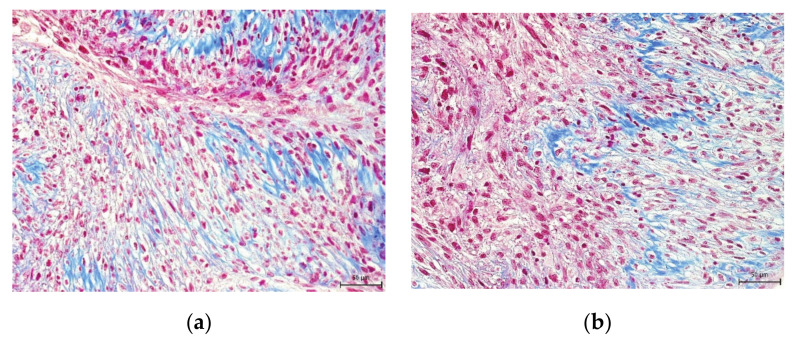
Detection of collagen and procollagen in the extracellular chondrogenic mass of the lineage Z01 after 3-week cultivation in chondrogenic differentiation medium. After histological staining using blue Masson’s trichrome, the collagen and procollagen appear as blue areas. Scale bar 50 µm. (**a**) DPSC-control; (**b**) DPSC-6M; (**c**) DPSC-12M; (**d**) non-differentiated cells.

**Figure 9 ijms-22-04432-f009:**
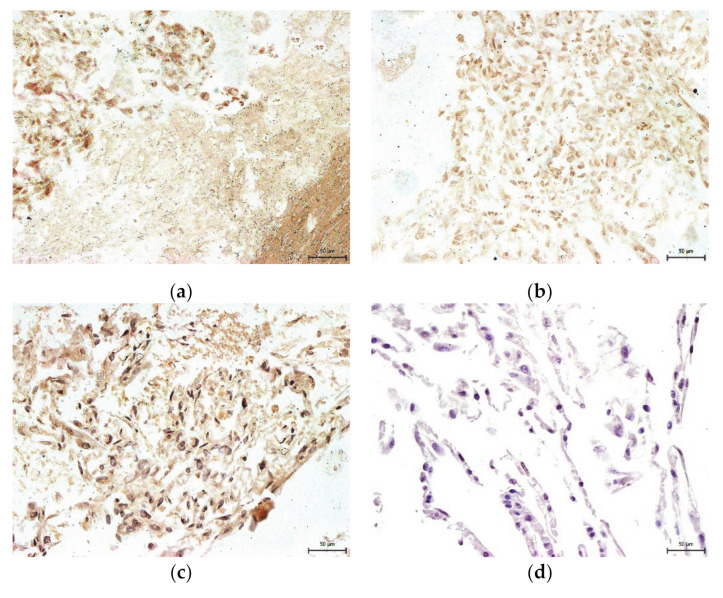
Immunocytochemical detection of osteocalcin, one of the main proteins of the extracellular osteogenic matrix. Samples of lineage Z08 are after 3-week cultivation in osteogenic differentiation medium. Osteocalcin is revealed as brown areas. Scale bar 50 µm. (**a**) DPSC-control; (**b**) DPSC-6M; (**c**) DPSC-12M; (**d**) non-differentiated cells.

**Figure 10 ijms-22-04432-f010:**
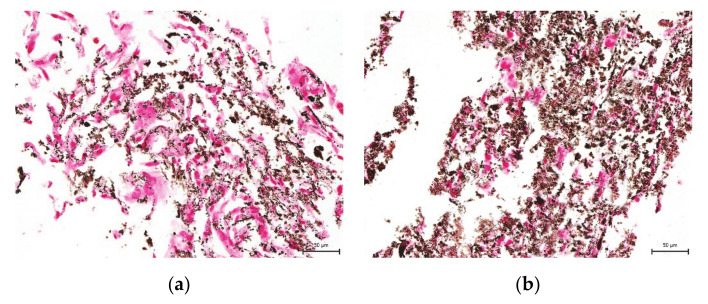
The presence of calcium phosphate deposits in the extracellular osteogenic matrix. Samples of lineage Z08 are after 3-week cultivation in osteogenic differentiation medium. Calcium phosphate deposits are colored as brown-black areas. Scale bar 50 µm. (**a**) DPSC-control; (**b**) DPSC-6M; (**c**) DPSC-12M; (**d**) non-differentiated cells.

**Figure 11 ijms-22-04432-f011:**
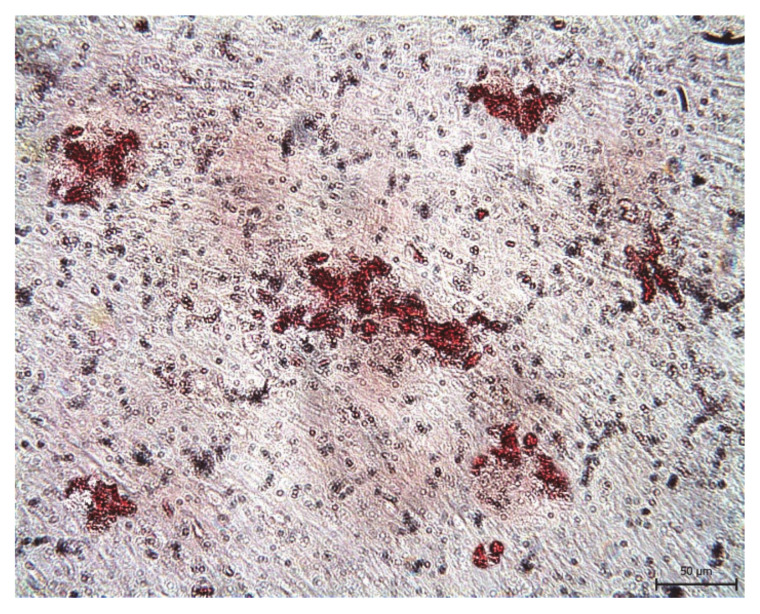
The visualization of adipose vacuoles and droplets in the extracellular matrix of non-cryopreserved DPSCs (Z02) after 3-weeks’ cultivation in adipogenic differentiation medium. After oil red staining, the adipose vacuoles are revealed as red areas, visible in an inverted optical microscope.

**Table 1 ijms-22-04432-t001:** Overview of donors’, sex, extracted teeth and their root development, and levels of eruption. (M—molar, P—premolars).

Lineage	Donors	Age	Tooth	Root Development	Eruption
Z01	Female	14	Mesiodens	Up to 1/2	Impacted
Z02	Female	16	M2	More than 1/2	Impacted
Z03	Female	16	M3	Up to 1/2	Impacted
Z04	Female	14	P1	Fully developed	Fully erupted
Z05	Female	15	M3	Up to 1/2	Impacted
Z06	Female	15	M3	Up to 1/2	Impacted
Z07	Female	18	M3	More than 1/2	Semi-impacted
Z08	Male	13	P1	More than 1/2	Impacted
Z09	Male	13	P1	More than 1/2	Impacted
Z10	Female	16	Supernumerary P	More than 1/2	Impacted

**Table 2 ijms-22-04432-t002:** Overview of analyzed CD markers.

CD Marker	Name	Cellular Expression	Function	DPSC Expression	References
CD10	Neprilysin: membrane metalloendopeptidase (MME), neutral endopeptidase (NEP)	CD10 is expressed on the B and T cell precursors, bone marrow stromal cells, lymphoblastic cells, Burkitt’s, and follicular germinal center lymphomas, immature B cells within the adult bone marrow, and on cells from patients with chronic myelocytic leukemia (CML). CD10 is also present on myoepithelial breast cells, bile canaliculi, fibroblasts, with incredibly high expression on the brush border of kidney and gut epithelial cells.Thought to be expressed during the first stages of heavy chain gene rearrangement and in an immunological context	It regulates B cell growth; it reduces the cellular response to peptide hormone in an immunological context.It is thought that the enzyme modulates the enkephalin-mediated inflammatory responseinvolved in the regulation of chemotactic and inflammatory processes involving neutrophils	Positive	[41,47,48]
CD13	Alanyl aminopeptidase (AAP) or aminopeptidase N (AP-N)	CD13 is expressed on the surface of early committed progenitors of granulocytes and monocytes CFU-GM, and all these lineages’ cells as they mature. It is also expressed on endothelial cells, epithelial cells from proximal renal tubules and intestinal brush border, bone marrow, stromal cells, fibroblasts, brain cells, osteoclasts, and cells lining bile duct canaliculi, and is expressed on a small proportion of large granular lymphocytes but not on other lymphocytes	It has a role in cell surface antigen presentation by trimming the N-terminal aa from MHC Class II-bound peptides. CD13 ectopeptidase activity is also thought to downregulate cellular responses to peptide hormones by reducing the local concentration of peptide available for receptor binding	Positive	[41,49]
CD18	Integrin beta chain-2: adhesive and signaling molecule for hematopoietic cell line	Lymphocyte Functions Associated Antigen 1 (binding CD18 and CD11) is a protein found on B cells, T cells, macrophages, neutrophils, and Natural Killer cells. The binding of CD18 and CD11b-d results in complement receptors’ formation, which are proteins mainly found on neutrophils, macrophages, and NK cells	The complement receptors participate in the innate immune response by recognizing foreign antigen peptides and phagocytizing them, thus destroying the antigen. The CD18 plays a role in the production and release of neutrophils from bone marrow	Negative	[50,51,52]
CD29	Integrin beta-1	Various cell types, including mesenchymal stem cells	Integrin family members are membrane receptors involved in cell adhesion and recognition in various processes, including embryogenesis, hemostasis, tissue repair, immune response, and metastatic diffusion of tumor cells	Positive	[41,53]
CD31	Platelet endothelial cell adhesion molecule (PECAM-1)	Human granulocytes, monocytes, and platelets	It plays a role in endothelial cell intercellular junctions. The encoded protein is a member of the immunoglobulin superfamily, and is likely involved in leukocyte transmigration, angiogenesis, and integrin activation	Negative	[54]
CD34	A transmembrane phosphoglycoprotein	Hematopoietic stem cells, hematopoietic progenitor cells, vascular endothelial progenitors, embryonic fibroblasts	CD34 plays a role in cell adhesion and the regulation of cell differentiation and proliferation	Negative	[55]
CD44	A cell-surface glycoprotein, Hermes antigen, Pgp-1	Leukocytes, erythrocytes; hematopoietic and non-hematopoietic cells (no platelets)	It binds hyaluronic acid, mediates adhesion of leukocytes, adhesion of leukocytes, endothelial cells, stromal cells, and ECM	Positive	[41,56]
CD45	Protein tyrosine phosphatase, receptor type C	It is present in various isoforms on all differentiated hematopoietic cells (except erythrocytes and plasma cells).	CD45 is an essential regulator of T- and B-cell antigen receptor signaling	Negative	[41,57]
CD49f	Alpha-6 Integrin	Memory T cells, thymocytes, monocytes, memory B cells, platelets, megakaryocytes, epithelial cells, endothelial cells, cytotrophoblasts	Alpha-6 Integrin, associates with CD29, binds laminin, adhesion and cell migration, embryogenesis	Positive	[58]
CD63	Granulophysin, LAMP-3 (Lysosomal-associated membrane 3), ME491, tetraspanin membrane protein	Several normal tissues, melanoma cells, a lysosomal membrane glycoprotein located mainly in the cytoplasm	It regulates melanoma cell motility and their adhesion and migration on substrates associated with beta1 integrins. It is associated with cell development and activation, and it may also function as a blood platelet activation marker	Positive	[59]
CD73	5’-nucleotidase, NTE, NT5	It is a surface enzyme expressed on multiple cells and used as a marker of lymphocyte differentiation	Enzyme catalyzes the conversion at neutral pH of purine 5-prime mononucleotides to nucleosides, the preferred substrate being AMP. It may also have a role in cell adhesion	Positive	[60]
CD90	Thy-1	It is expressed by hematopoietic stem cells, and neurons in all species studied, and it is highly expressed in connective tissue and various fibroblast and stromal cell lines. It is also expressed on all thymocytes and peripheral T cells in mice, but, in humans, it is expressed only on a small % of fetal thymocytes, 10–40% of CD34+ cells in bone marrow, and <1% of CD3+CD4+ lymphocytes in the peripheral circulation. It is also expressed by human lymph node HEV endothelium but not other endothelia	Thy-1 can be used as a marker for a variety of stem cells and the axonal processes of mature neurons	Positive	[41,61,62]
CD105	EndoglinSH2	Endothelial cell, mesenchymal stem cell, erythroid precursors, activated monocytes, macrophages, and human vascular endothelial cells. Expression of endoglin is elevated on the endothelial cells of healing wounds, developing embryos, inflammatory tissues, and solid tumors.Endoglin is a marker of activated endothelium, and its vascular expression is limited to proliferating cells.	The possible ligand for an integrin, angiogenesis, modulates response to TGF beta 1postulated that endoglin is involved in the cytoskeletal organization affecting cell morphology and migrationdevelopment of the cardiovascular system, and in vascular remodelingprotective role of CD105 against pro-apoptotic factors	Positive	[41,63,64]
CD106	Vascular cell adhesion protein 1 (VCAM-1)	Activated endothelial cellsmyeloid lineage and bone marrow stromal cellsfollicular dendritic cells, interdigitating reticulum cells, and Kupffer cells	It is an adhesion molecule, ligand for VLA-4, leukocyte adhesion, transmigration, and T cells’ co-stimulation.It encodes a cell surface sialoglycoprotein expressed by cytokine-activated endothelium. This type I membrane protein mediates leukocyte-endothelial cell adhesion and signal transduction, and may play a role in developing atherosclerosis and rheumatoid arthritis.CD106 expressed in non-vascular tissues has been implicated in the interaction of hematopoietic progenitors with bone marrow stromal cells, B cell binding to follicular dendritic cells, co-stimulation of T cells, and embryonic development	Positive	[41,65,66]
CD117	Tyrosine-protein kinase KIT, mast or stem cell growth factor receptor (SCF)	KIT is a cytokine receptor expressed on the surface of hematopoietic stem cells and other cell types	Signaling through KIT plays a role in cell survival, proliferation, and differentiation. For instance, KIT signaling is required for melanocyte survival, and it is also involved in hematopoiesis and gametogenesis	Positive	[45,46]
CD146	Melanoma cell adhesion molecule (MCAM), cell surface glycoprotein (MUC18)	Activated human T cells, endothelial progenitors such as angioblasts and mesenchymal stem cells, and strongly expressed blood vessel endothelium and smooth muscle	CD146 has been seen as a marker for mesenchymal stem cells isolated from multiple adult and fetal organs, and its expression may be linked to multipotency; mesenchymal stem cells with greater differentiation potential express higher levels of CD146 on the cell surface	Positive	[41,67,68]
CD166	Activated leukocyte cell adhesion molecule (ALCAM)	T cells, NK cells, platelets, thymocytes, activated B and T cells, eosinophils, fibroblasts, endothelial cells, keratinocytes, monocytes, epithelial cells, neurons, MSCs	Adhesion molecule, ligand for CD6; T-cell activation. Involved in neurite extension by neurons via heterophilic and homophilic interactions	Positive	[41,69,70]
CD 271	Low-affinity nerve growth factor receptor (LNGFR), p75 neurotrophin receptor	This receptor is currently considered one of the most efficient markers for prospectively isolate BM-MSCs and other adult stem cells	It is a common receptor of neurotrophins that mediates different biological effects in several cell types, like cell survival, apoptosis, migration, and differentiation, and neurotrophin binding, interacting transmembrane co-receptors expression, intracellular adaptor molecule availability, and post-translational modifications, such as regulated proteolytic processing	Positive	[71]
MHC class I	Major histocompatibility complex (MHC) class I	MHC molecules are found on the cell surface of all nucleated cells in vertebrates	Major histocompatibility complex (MHC) class I and class II proteins play a pivotal role in the immune system’s adaptive branch. Both classes of proteins share the task of presenting peptides on the cell surface for recognition by T cells	Positive	[72]
MHC class II	Major histocompatibility complex (MHC) class I	MHC class II molecules are found only on professional antigen-presenting cells such as dendritic cells, mononuclear phagocytes, some endothelial cells, thymic epithelial cells, and B cells		Negative	[73]
STRO1	A gene for a protein marker of mesenchymal stem cells (MSC)	Marker for mesenchymal stem cells.	A gene for a protein marker of mesenchymal stem cells (MSC)	Positive	[41,74]

## Data Availability

Data is contained within the article.

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
