# Peer review of "The Effects of Cryogenic Storage on Human Dental Pulp Stem Cells"

_ijms, 2021, doi:10.3390/ijms22094432_

Round 1

Reviewer 1 Report

The present paper is very interesting and perforormed in a rigprous way. Authors showed the comparison in terms of  cell functions and morphology after cryostorage at -80°C of dental pulp stem cells.

To better evalute and demontsrate the ability to differentiate into osteogenic, chondrogenic and adipogenic lineage authors shuld add experiments as WB or RT-PCR to demonstrate the expression of specific markers of different commitment as FABP4, OPN, RUNX2, Col1, ACAN, Col2, PPARg....or others

Reviewer 2 Report

The presented study is focused on the possible effect of uncontrolled-rate freezing method on the recovery and function of dental pulp stem cells (DPSCs). The authors confirmed that DPSCs stored for 6 or even 12 months maintained their stemness, proliferation and differentiation capacity.

The paper is very interesting and well written. It brought significant biological findings about the used cryopreservation method, which might be possibly applied also for other types of adult stem cells and thus be very useful for regenerative medicine.

Below are listed minor comments that should be corrected and clarified before the final acceptance of paper:

Introduction:

Lines 88-93: Authors mentioned only few studies with DPSCs in Introduction and one study in Discussion (Lines 260-261) in which uncontrolled-rate freezing method was used to cryopreserve stem cells. This rising a question, whether this method is suitable also for other types of mesenchymal stem cells. Or this method is suitable only for DPSCs? This could be clarified within the text.

Materials and methods:

Lines 375-378: The description of uncontrolled-rate freezing is little bit unclear.  A term “first” is repeated within the text. However, it is not clear if the CP (DMSO) was cooled down at first, then it was mixed with medium and FBS and finally mixed with cell pellet. Or the cryopreservation medium composed of FBS and 10% of DMSO was cooled down to 4 °C and then mixed with cell pellet, which is usually the most common method. Please correct the right meaning.

Line 383: Did you thaw one cryovial form each sample (lineage). If yes, please include this information within the text. If not, specify how many different samples (cryovials) were thawed and used for subsequent analysis.

Line 395: Please correct: “We assessed the cell viability using a trypan…”

Lines 398-399: Please describe briefly or cite how you observe the population doublings and population doubling time of cultured DPSCs. Is it known how many passages can be obtain from human DPSCs before their loss the proliferative capacity and start to differentiate? This could be an interesting fact, which might be mentioned in the discussion.

Lines 406-407: Please provide the information about antibodies used in this study (clone, isotype, fluorochrome, producer etc.) in order to make the staining procedure clear. The information should be included within the text in Methodology section, or as new table or as an appendix at the end of paper. This can ensure the reproducibility of the obtained results.

Lines 437-439: Did you use both media, first for induction and second for maintaining the culture? Or you used one medium for some samples and the second one for other samples? Please clarify it, and omit using "either, or" phrase if both media were used subsequently.

Results:

Lines 124-130: Why did you not analyse the cell viability immediately after thawing? Testing the cell viability of the 2nd passage indicated a very good recovery of the cells after thawing, however, it does not reflect the possible effect of the cryopreservation on the viability of thawed cells. Therefore, it is strongly recommended to include this analysis in the future studies. Also, the conclusions should be slightly changed for example: “The cryopreservation did not affect the recovery of thawed DPSCs and the viability of subsequent passages.”

Lines 143-144: “We also observed a statistically significant difference in the 5p and 7p.” This does not agree with the Figure 4, in which a statistically significant difference is indicated for 6p!!! Please correct the text or figure.

Lines 167-180: Expression of several analysed markers, either positive or negative, is not mentioned within the results (CD10, CD18, CD63, CD105, CD271, MHCI, MHCII and STRO1). Their expression should be described even tangentially, so the reader can check the specific expression in Figure 6.

Discussion:

Lines 278-280: If some cells are lysed or disrupted due to the freezing or thawing procedure, then a control of cell viability immediately after thawing is desired as was mentioned before.

Line 281: Please correct “cell depth” to cell death”.

Lines 307-309: The change of stem cell phenotype after cryopreservation was also observed and discussed in the study of Kulikova et al. (2019): Survivability of rabbit amniotic fluid-derived stem cells post slow-freezing or vitrification. This might support your findings.  

References:

The title of Ref. 8 should be in lower case.

Round 2

Reviewer 1 Report

NA